# Goss’s Wilt Resistance in Corn Is Mediated via Salicylic Acid and Programmed Cell Death but Not Jasmonic Acid Pathways

**DOI:** 10.3390/plants12071475

**Published:** 2023-03-28

**Authors:** Alexander Shumilak, Mohamed El-Shetehy, Atta Soliman, James T. Tambong, Fouad Daayf

**Affiliations:** 1Department of Plant Science, University of Manitoba, Winnipeg, MB R3T 2N2, Canada; 2Department of Botany, Faculty of Science, Tanta University, Tanta 31527, Egypt; 3Department of Genetics, Faculty of Agriculture, University of Tanta, Tanta 31527, Egypt; 4Agriculture and Agri-Food Canada, Ottawa, ON K1A 0C6, Canada

**Keywords:** *Clavibacter nebraskensis*, maize, jasmonic acid, salicylic acid, programmed cell death

## Abstract

A highly aggressive strain (CMN14-5-1) of *Clavibacter nebraskensis* bacteria, which causes Goss’s wilt in corn, induced severe symptoms in a susceptible corn line (CO447), resulting in water-soaked lesions followed by necrosis within a few days. A tolerant line (CO450) inoculated with the same strain exhibited only mild symptoms such as chlorosis, freckling, and necrosis that did not progress after the first six days following infection. Both lesion length and disease severity were measured using the area under the disease progression curve (AUDPC), and significant differences were found between treatments. We analyzed the expression of key genes related to plant defense in both corn lines challenged with the CMN14-5-1 strain. Allene oxide synthase (*ZmAOS*), a gene responsible for the production of jasmonic acid (JA), was induced in the CO447 line in response to CMN14-5-1. Following inoculation with CMN14-5-1, the CO450 line demonstrated a higher expression of salicylic acid (SA)-related genes, *ZmPAL* and *ZmPR-1*, compared to the CO447 line. In the CO450 line, four genes related to programmed cell death (PCD) were upregulated: respiratory burst oxidase homolog protein D (*ZmrbohD*), polyphenol oxidase (*ZmPPO1*), ras-related protein 7 (*ZmRab7*), and peptidyl-prolyl cis-trans isomerase (*ZmPPI*). The differential gene expression in response to CMN14-5-1 between the two corn lines provided an indication that SA and PCD are involved in the regulation of corn defense responses against Goss’s wilt disease, whereas JA may be contributing to disease susceptibility.

## 1. Introduction

Routinely in the top three most-produced crops globally, corn is an increasingly important cash crop in Canada, with the country contributing 1.2% of the globe’s overall production [1]. With the increased demand for corn and corn products, there is an increased demand for more corn acres. This leads to tighter corn rotations and an increase in the incidence of corn diseases such as Goss’s wilt, which is triggered by the bacterial pathogen *Clavibacter nebraskensis* (*Cn*) [2]. This highly damaging Gram-positive bacterium grows as orange-colored colonies in agar culture media [3,4] and is part of a super-group of bacteria within the genus Clavibacter that infect a variety of other crops [3,5,6,7]. Goss’s wilt has reached every state in the U.S. corn belt and every Canadian province with a significant corn production, and it has been on the rise since the adoption of glyphosate as the main method of weed control instead of conventional tillage [2,8,9,10,11]. Depending on where *Cn* enters the plant, it can cause either corn wilt or blight [8,9,12,13]. The bacterium primarily resides in the harvest residues of previous corn crops and enters new plants via wounds and roots [8,14]. The most common symptoms are dark green water-soaked lesions that dry out and form scorched-looking lesions over time with disease progression and wind currents [8,14,15,16]. In severe cases, yield reductions as high as 50% have been reported. Since there is no known effective chemical control for *Cn*, producers rely on tillage, clean farm equipment, and primarily good corn genetics [9,14,17,18,19,20,21]. Controlling alternative hosts such as *Seteria viridis* (green foxtail) may also reduce the *Cn* transmission to and infection of corn plants [17,22]. 

Given that the manipulation of Gram-positive bacteria is difficult, there is a lack of sufficient information about the functional genetic makeup of *Cn* [23]. *Cn* can colonize corn tissues using a type II secretion system to transfer virulence factors such as proteases, cellulases, chitanases, and β-1,4-xylanases [4,24,25]. Additionally, few details are available about the mechanisms that *Cn* employs to invade corn tissues or how to counteract the disease progression of Goss’s wilt. The hypersensitive response (HR) is a prompt cell reaction to foreign attacking organisms and is achieved by activating molecular systems that end disease progression [26,27]. Programmed cell death (PCD) refers to the process of rapid cell death that occurs at the infection site and is considered a component of HR. In theory, PCD should prevent the progression of disease since there is no more live tissue around the pathogen to infect. Nitric oxide and reactive oxygen species (ROS) are key elements of PCD, as they can kill the infected plant cells [27,28]. The plant NADPH/respiratory burst oxidase D (*RbohD*) gene is a key player in the generation of ROS, such as hydrogen peroxide (H_2_O_2_) and superoxide, that kill the cells at the infection site. *RbohD* was upregulated in corn at the site of infection by *Cn*, suggesting a potential role in corn defense against this bacterium [28]. Different approaches, including an expression quantitative trait locus (QTL) analysis and the transcription profiling of resistant and susceptible corn genotypes, revealed complex molecular plant–pathogen interactions, including shifts not only in the expression of genes linked to defense responses mediated by salicylic acid (SA) and jasmonic acid (JA) but also in the oxidative status of infected tissues [29,30]. 

For SA, we tested two genes, *ZmPAL and ZmPR-1*, which are known to play important roles in the SA defense pathway against biotrophic pathogens. For PCD, we tested NADPH oxidase gene *RbohD*, a crucial mediator in ROS production, as well as polyphenol oxidase (*PPO1*), *ZmRab7*, and *ZmPPI*. For JA, we assayed the transcript levels of six selected genes: *ZmAOS*, *ZmAOC1*, *ZmLox9*, *ZmJaz12*, *ZmMYC7*, and *ZMERF147*. Oxylipins, including JA, are lipid-derived signaling molecules that participate in a wide variety of developmental processes and play roles in mediating defense responses to biotic and abiotic stress in plants [31]. The oxylipin biosynthesis begins with the oxidation of polyunsaturated fatty acids to form fatty acid hydroperoxides via enzymatic peroxidation catalyzed by lipoxygenases (LOXs) [32]. In maize, six genes are predicted to encode 13-lipoxygenases (*LOX7*, *LOX8*, *LOX9*, *LOX10*, *LOX11*, and *LOX13)*, and seven genes encode 9-lipoxygenases (*LOX1*, *LOX2*, *LOX3*, *LOX4*, *LOX5*, *LOX6*, and *LOX12*), which convert 18:3 α-linolenic acid and 18:2 α-linoleic acid to 10-oxo-11-phytodienoic acid (10-OPDA) and 10-oxo-11-phytoenoic acid (10-OPEA), respectively [33]. Upon oxygenation by a 13-lipoxygenase, an allene oxide is formed by allene oxide synthase (AOS) and is subsequently cyclized by an allene oxide cyclase (AOC) to OPDA [34]. The expression of key biosynthetic marker genes of the JA signaling pathway, *LOX2*, *LOX3*, *AOC1* (allene oxide cyclase), and *AOS*, in *ZmGLP1* (a Germin-like Protein from Maize)-overexpressing Arabidopsis was strongly induced after *Pst*DC3000 and *Sclerotinia sclerotiorum* infection [35]. *ZmLOX*9 was selected based on previous studies that showed that LOX9, belonging to the 13-lipoxygenases family, plays an important role in defense against *Bipolaris maydis*, which is responsible for causing southern leaf blight in corn and is also involved in JA biosynthetic pathways [36,37]. Jaz (Jasmonate ZIM-domain) family proteins serve as transcriptional repressors of the JA signaling pathway, preventing the plant from being overwhelmed by the overactivation of the pathway, which causes unintended plant damage [38,39,40,41,42]. ZmMYC7 is a putative MYC2 ortholog that plays a crucial role in protecting maize against *Fusarium graminearum* via the JA signaling pathway. ZmMYC7 was found to bind to G-box cis-elements in the *ZmERF147* promoter in vitro and activate its transcription. However, this activation was impeded by two other proteins, ZmJAZ11 and ZmJAZ12 [43].

Significant efforts to decipher *Cn*–corn interactions have been made in order to develop better control strategies and prevent outbreaks. In this article, we aim to determine the role of SA and PCD defense-related genes in corn–*Cn* interactions. Our results indicate potential roles for the SA pathway and PCD in corn defense against the bacterial pathogen *Cn*, whereas the JA pathway showed little involvement in the successful reduction of Goss’s wilt disease in corn.

## 2. Results

### 2.1. Pathogenicity

To understand how corn plants respond to Goss’s wilt at the molecular level, corn lines that are tolerant (CO450) and susceptible (CO447) to Goss’s wilt were inoculated with CMN14-5-1. Relative to the uninoculated corn plants and *Cn*-inoculated CO450 corn lines, lesions in the CO447 lines spread rapidly parallel to the leaf veins (Figure 1A). Figure 1B shows the total AUDPC in CO450 and CO447 plants inoculated with CMN14-5-1. No AUDPC was calculated for the control wounded and unwounded plants because the lesions from the initial wounded areas did not progress. The inoculated CO450 and CO447 plants showed a strong significant difference. The CO450 corn line had a significantly lower total AUDPC than CO447 at the end of the experiment, whereas no difference was observed between the wounded controls in both tested lines. The disease severity increased over time in the corn lines both tolerant and susceptible to CMN14-5-1. Figure 1C shows a clear difference in both inoculated corn lines in which the disease severity levels were strongly induced in the susceptible CO447 in comparison with the tolerant CO450. 

### 2.2. Plant Defense against Goss’s Wilt Is Not Enhanced via the Jasmonic Acid Pathway

The enhanced disease resistance of the CO450 (tolerant) corn plants against the aggressive strain, CMN14-5-1, led us to further research to assay the transcript levels of genes related to plant defense using a reverse transcriptase qPCR. Treatment with CMN14-5-1 caused a 13.7-fold increase in *ZmAOS* expression in CO447 and a minor increase (2.9-fold) in CO450 at 2 dpi compared with 0 dpi (Figure 2A). Additionally, the *ZmAOS* expression in CO447 (susceptible) increased by 10.8-fold in comparison with CO450 at 2 dpi. On the other hand, treatment with CMN14-5-1 did not cause any significant changes in *ZmAOC1* (Figure 2B), and *ZmLOX9* (Figure 2C) in the CO450 lines. In addition, the transcript levels of *ZmAOC1* (Figure 2B) were repressed in CO447 lines at 2 dpi. In contrast, *ZmLOX9* (Figure 2C) showed an induction of 0.9-fold (2.5 times) in CO447 at 2 dpi compared to 0 dpi. Treatment with CMN14-5-1 led to a 1.6-fold (7 times) increase in the transcript abundance of *ZmJaz12* in CO450 at 2 dpi compared with 0 dpi but did not lead to an increase in this gene’s transcripts in CO447 (Figure 2D). Furthermore, *ZmMYC7* (Figure 2E), and *ZMERF147* (Figure 2F) were significantly downregulated in CO447 and CO450 lines at 2 dpi compared to 0 dpi.

### 2.3. SA and PCD Regulate Goss’s Wilt Disease Resistance 

The plant hormone SA is a key player in plant defense. To further study the correlation between the disease resistance and genes associated with the defense mechanisms to CMN14-5-1, we measured the expression levels of Phenylalanine ammonia-lyase (*PAL*), which is a key gene for pathogen-induced SA accumulation in plants, and the SA-responsive marker genes pathogenesis-related proteins-1 (*PR-1*) in both tested corn lines. According to our data, the expression levels of *ZmPAL* and *ZmPR-1* were strongly induced at 2 dpi in the tolerant CO450 line, with a 1.6-fold (2.6 times) and 7.8-fold (2.1 times) enhancement, respectively, when compared to the susceptible CO447 line (Figure 3A,B). Inoculation with *Cn* did not induce high enough levels of *ZmPAL* and caused an increase of *ZmPR-1* (6.1-fold; 7.3 times) in CO447 at 2 dpi in comparison with 0 dpi, whereas pathogen infection in the CO450 tolerant line induced significantly higher levels of *ZmPAL* (2-fold; 4.4 times) and *ZmPR-1* (13.7-fold; 11.4 times) in the CO450 tolerant line at 2 dpi in comparison with 0 dpi (Figure 3A,B). Figure 3C shows the higher gene induction of respiratory burst oxidase D (*ZmRbohD*) within the tolerant CO450 line compared to the susceptible CO447 line at 0 and 2 dpi, as the differences between the two lines were 37 and 14 times, respectively. Although CMN14-5-1 inoculation induced significantly higher levels of polyphenol oxidase (*ZmPPO1*) at 2 dpi in both the susceptible CO447 (2.2 times) and tolerant CO450 (2.6 times) corn lines, there were significant variations in *ZmPPO1* expression levels between the two tested corn lines at 0 and 2 dpi in which the tolerant CO450 line showed a change of 3.4 and 4 times over the susceptible CO447 line, respectively (Figure 3D). We also assayed the *ZmRab7* gene in the corn lines both tolerant and susceptible to CMN14-5-1. At 0 and 2 dpi, the transcript abundance of the *ZmRab7* gene in the tolerant CO450 line was roughly two times higher than that in the susceptible CO447 line (Figure 3E). Figure 3F displays the response of the peptidyl-prolyl cis-trans isomerase (*ZmPPI*) gene in the tested corn plants inoculated with CMN14-5-1. At 0 and 2 dpi, the tolerant CO450 line displayed levels of *ZmPPI* gene expression that were 4.3 and 11.1 times higher than the susceptible line of corn (Figure 4A).

### 2.4. Exogenous Application of SA and H_2_O_2_ Confers Partial Disease Resistance against CMN14-5-1

As a next step, we tested whether the application of synthetic SA or H_2_O_2_ could induce disease resistance in the susceptible cultivar CO447 by spraying the corn plants 48 h before inoculation. Interestingly, both SA and H_2_O_2_ were able to confer partial resistance in CO447 against the aggressive strain CMN14-5-1 compared with the untreated inoculated plants (Figure 4A–C). The AUDPC values for both lesion length and disease severity in SA- and H_2_O_2_-treated CO447 corn plants in response to CMN14-5-1 were significantly lower than those in untreated CO447 corn plants. The AUDPC values for lesion length were 1.66 and 1.5 times lower for SA- and H_2_O_2_-treated CO447 plants, respectively, than for untreated plants (Figure 4E). Furthermore, the AUDPC values for leaf disease severity were 1.54 and 1.62 times lower for SA- and H_2_O_2_-treated CO447 plants, respectively, in comparison to the untreated CO447 plants (Figure 4F).

## 3. Discussion

This study illustrates that corn resistance against Goss’s wilt disease is promoted through SA and programmed cell death. Significant differences were shown between the two tested corn lines, CO447 and CO450, in response to the highly aggressive *Cn* bacterial strain CMN14-5-1. The disease severity was higher in the susceptible CO447 line in comparison with the tolerant CO450 line, and this was represented by the highly significant difference in the length of lesions. To better understand the involvement of corn defense genes against *Cn*, we conducted gene expression experiments using qRT-PCR analysis to compare the relative expression of uninoculated and inoculated corn lines. 

Given that most plant defenses are associated with either JA or SA signaling pathways, we tested well-studied marker genes for each pathway. To test if the defense responses induced in the CO450 line were associated with the JA signaling pathway, we assayed six genes that are known to be part of JA biosynthesis and signaling defense pathways. JA has been shown to respond to both biotic and abiotic stresses [44]. Thus, we analyzed the expression levels of *ZmAOS* (Figure 2A), *ZmAOC1* (Figure 2B), *ZmLox9* (Figure 2C), *ZmJaz12* (Figure 2D), *ZmMYC7* (Figure 2E), and *ZmERF147* (Figure 2F). The susceptible CO447 line had a higher-fold increase in *ZmAOS* transcript abundance than the resistant CO450 line. Additionally, inoculation with *Cn* caused a prominent induction in the expression of *ZmJaz12*, the JA transcriptional repressor gene, in CO450 compared with CO447. There was also no significant induction of *ZmAOC1* and *ZmLOX9* in the tolerant CO450 line. In addition, a significant repression in the transcript levels of *ZmMYC7* and *ZMERF147* was found in both tested corn lines. Together, these results suggest that disease resistance against CMN14-5-1 was independent of the JA defense signaling pathway. 

The other important signaling molecule is SA, which has key regulatory functions in plant defense and plant development [45,46,47]. We tested two genes known to play important roles in the SA defense pathway against biotrophic pathogens, *ZmPAL* and *ZmPR-1*. CMN14-5-1 induced the expression levels of *ZmPAL* and *ZmPR-1* at a higher rate in CO450 compared to the CO447 plants. Generally, a reduction in *PAL* activity makes plants more vulnerable to disease because SA accumulation is reduced and systemic acquired resistance is abolished [47]. Both *PAL* and SA contribute to corn defense against the sugarcane mosaic virus infection [47]. It has also been reported that *PR-1* genes have the capacity to inhibit PCD within the initial lesion caused by *Pseudomonas syringae pv. tabaci* and eventually cause plant death after the pathogen is contained by PCD [48]. In addition, previous studies have demonstrated that *PR-1* genes possess antibacterial properties effective against both Gram-positive and Gram-negative bacteria [49]. This data prompted us to investigate PCD-related genes. Following pathogen colonization, ROS can either strengthen the cross-linking of the plant cell walls or promote the oxidative burst to effectively collapse the invaded plant tissues, resulting in limiting the disease spread within the plant [28,50,51,52,53]. ROS production is closely linked to PCD in response to biotic stresses [28]. ROS are produced in animals and plants by the NADPH oxidase family of enzymes [54]. Plant NADPH oxidases are commonly referred to as respiratory burst oxidases (Rboh) due to their closest functional similarity to mammalian NADPH oxidases [55]. NADPH oxidase/*RbohD* is a crucial mediator in ROS production and the activation of PCD in many plants [28,50,52,56,57]. Therefore, we tested *RbohD*, which is a NADPH oxidase involved in ROS production and PCD activation [57]. We also quantified polyphenol oxidase (*PPO1*), which plays a key role in plant defense by using molecular oxygen to convert *ortho-*diphenols into *ortho*-quinones. The *ortho-*quinones are part of the browning reaction correlated with tissue damage, which has been suggested to restrict pathogen progression as a small HR [51,58,59]. The induction of *PPO1* gene expression in tomato plants has been reported to trigger resistance against *P. syringae pv. tomato* [59]. Since *Rab7* plays a critical role in regulating ROS scavengers to protect plants from mainly abiotic stresses and in the conversion of phagosomes into lysosomes, which deactivate potentially toxic secretions in response to pathogen infection [60,61,62], we assayed *Rab7.* The induced expression levels of *Rab7* in wheat plants in response to *Pucinia striiformis f.* sp. *Tritici* had suggested the regulatory role of *Rab7* following pathogen colonization to prevent disease spread [61]. Furthermore, we tested the peptidyl-prolyl cis-trans isomerase (*PPI*) gene, which has been reported to play various roles in plants, including protein folding, stress response, plant development, and redox reaction regulation [63]. When a pathogen attacks a plant, it triggers the HR response, which relies heavily on the ROS that are produced through redox reactions. These reactions must be tightly controlled to protect the plant from additional damage resulting from excess ROS. In order to limit the harmful impact caused by ROS, plants activate the PPI gene. Moreover, the PPI gene is expressed at a relatively higher level than that of the control treatment, and the early induction of *RbohD* transcripts during the infection might indicate that the *RbohD* gene is regulated by the *PPI* gene [63]. The *PPI* and *Rab7* genes are strongly linked to *RbohD* downregulation by producing ROS scavengers that eliminate the harmful impacts of excess ROS, protecting plants from further damage [60,62,63,64]. Our findings revealed an increase in the transcript abundance of *ZmRbohD*, *ZmPPO1*, *ZmRab7*, and *ZmPPI* at 0 and 2 dpi time points in CO450 compared to CO447, implying a critical role for PCD-related genes in inducing plant defense against Goss’s wilt disease. Together, these results strongly imply that SA and PCD are essential in regulating defense responses against Goss’s wilt disease in corn. 

SA and H_2_O_2_ treatments induce disease resistance and systemic acquired resistance in plants [65,66,67,68]. SA confers disease resistance against the downy mildew pathogen, *Peronosclerospora maydis*, by promoting the expression levels of *PR-1* and *PR-5* genes [69]. It has also been shown that SA triggers PR genes in rice and barley [70,71]. Additionally, the synthetic chemical analogue of SA, Benzothiadiazole (BTH), was shown to activate an enhanced resistance in wheat to powdery mildew caused by *Erisyphe graminis*, the leaf-rust-causing *Puccinia recondita*, and Septoria leaf spot. *Magnaporthe grisea*, the bacterium that causes rice blast, was also reported to be controlled by the BTH treatment of rice seedlings [72]. Consistent with these results, the exogenous application of SA or H_2_O_2_ was able to restore partial disease resistance in the susceptible CO447 plants against CMN14-5-1. These results suggest that SA and H_2_O_2_, in combination with other important components, likely act in an additive manner to induce disease resistance against Goss’s wilt.

## 4. Materials and Methods

### 4.1. Plant Material

The corn (*Zea mays* L.) lines utilized in this research, namely, CO447 and CO450, were provided by Dr. Lana Reid from Agriculture and Agri-Food Canada located in Ottawa, Ontario. These inbred lines do not have common parental ancestry. Dr. Reid requested that Dr. Daayf’s lab test the two lines in the field, and the results showed that CO447 was susceptible to *Cn* infection while the other was tolerant [73]. The plants were grown in a controlled environment with a 16/8 h light/dark cycle and a temperature of 22/18 °C for the day/night cycle.

### 4.2. Chemical Treatments 

SA (Sigma; Steinheim, Germany; cat. no. 247588) and H_2_O_2_ (Sigma; St. Louis, MO 63103, USA; cat. no. 216763) treatments were carried out using 500 μM solutions. SA was made ready for use by diluting it in water. The H_2_O_2_, which was available as a 30% solution, was also diluted in water. Each of the dilutions was freshly made. Treatments were carried out by spraying solutions on three- to four-week-old maize plants 48 h prior to bacterial inoculation. 

### 4.3. Bacterial Isolation, Preparation of Inoculum, and Leaf Inoculation 

The highly aggressive CMN14-5-1 [13] was grown for 2–3 days at 23 °C in a nutrient broth yeast medium (NBY) containing 8 g/L nutrient broth (Becton, Dickinson and Company; Sparks, MD 21152, USA; cat. no. DF0003178), 2 g/L yeast extract (Becton, Dickinson and Company; Sparks, MD 21152, USA; cat. no. B11929), 15 g/L agar (Becton, Dickinson and Company; Sparks, MD 21152, USA; cat. no. DF0812179), 2 g/L K_2_HPO_4_ (Fisher; Ottawa, Canada; cat. no. P288-500), 0.5 g/L KH_2_PO_4_ (Fisher; NJ 07410, USA; cat. no. P-284, CAS-7778-77-0), 5 g/L glucose (Sigma; St. Louis, MO 63103, USA; cat. no. G8270), and 0.246 g/L MgSO_4_·7H_2_O (Fisher; New Jersey 07410, USA; cat. no. M-67, CAS-7487-88-9) [74]. The inoculum consisted of bacterial cells that were mixed with a phosphate buffer solution containing 10 mM of both monobasic potassium phosphate and dibasic potassium phosphate at pH 6.7. The bacterial culture concentration was measured and then adjusted to 1 × 10^7^ CFU/mL for the inoculation process [15]. 

Maize plants were mechanically wounded at the V5 leaf stage using a disposable syringe plunger covered with a 5 mm sandpaper disc. The third, fourth, and fifth leaves were wounded on both sides of the midrib [13]. The control plants had 20 µL of phosphate buffer solution applied to their wounds, while the inoculated plants received 20 µL of CMN14-5-1 inoculum. For the two types of corn tested, three different treatments were used in the experiment. The first treatment was an unwounded control, while the second treatment involved wounding the corn leaves and then treating them with a phosphate buffer. The third treatment also involved wounding the leaves, but this time they were treated with CMN14-5-1. The phosphate buffer did not have any negative effects on the control treatments. Experiments were carried out in three biological replicates, with each replicate consisting of six plants. After being treated, the plants were kept in a mist chamber overnight with 100% relative humidity before being moved to a growth room for a period of 10 days.

### 4.4. Measurement of Lesion Length, Disease Severity Rating, and Sampling 

Treatments and time intervals were assigned to each inbred corn line. This experiment had six time intervals: 0 (15 min after inoculation), 2, 4, 6, 8, and 10 dpi. Three different treatments were used, including: (1) no wound (control); (2) wound with PPB (control); and (3) wound with *CMN14-5-1*. The size of the lesion was measured in both directions from the infection site at each specific time point. For each replicate, the AUDPC was calculated using the mean of six biological replicates comprising six subsamples. The disease severity index was used to assess disease severity with the following scale: 0, the only lesion is the initial wound; 1, chlorosis or reddening only; 2, chlorosis or reddening accompanied by freckling and approximately 10% necrosis; 3, chlorosis or reddening accompanied by freckling and approximately 11–25% necrosis or wilting; 4, 26–50% necrosis; 5, 51–75% necrosis; and 6, 76–100% necrosis [13]. Leaf segments that contained the entire inoculation site plus the diseased area were excised and analyzed at 0 (15 min post inoculation) and 2 dpi for transcript analysis. 

### 4.5. Measurement of Transcript Levels

TRI Reagent (Invitrogen; Vilnius, Lithuania; cat. no. AM9738) was used to extract the total RNA, which was then treated with DNase I (ThermoFisher Scientific; Vilnius, Lithuania; cat. no. EN0521) and reverse-transcribed into cDNA using the RevertAid First Strand cDNA Synthesis Kit (ThermoFisher Scientific; Vilnius, Lithuania; cat. no. K1622). Appendix A lists all the primers used in gene expression studies. The 2^−ΔΔCT^ method was used to quantify the relative gene expression [75], with actin serving as the reference gene. 

### 4.6. Statistical Analysis 

Statistical analyses for the experiments in Figure 1, Figure 2 and Figure 3 were carried out using the PROC MIXED function of Statistical Analysis Software (SAS) (Release 9.1 for Windows; SAS Institute, Cary, NC, USA). The Tukey test (α = 0.05) was used to compare treatment means for the experiments in Figure 4. Each experiment was repeated three times, with six plants in each replicate.

## Figures and Tables

**Figure 1 plants-12-01475-f001:**
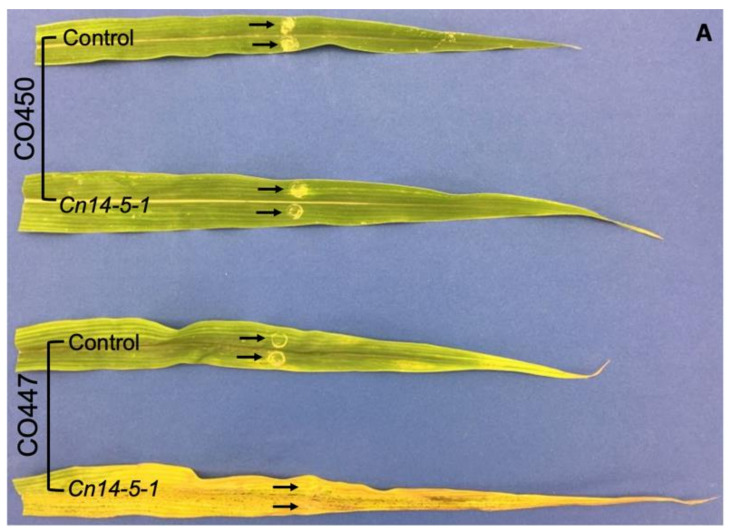
Increased disease resistance to *C. nebraskensis* (CMN14-5-1) in the tolerant CO450 corn line. (**A**) Phenotype of the tested corn plants at 14 dpi (days post inoculation). Wounded susceptible CO447 and tolerant CO450 plants were pretreated with either the phosphate buffer (PPB; control) or CMN14-5-1 (a highly aggressive *Cn* strain). Control and inoculated leaves were treated with 20 μL of buffer or bacterial suspension, respectively. Arrows point to the inoculation site. (**B**) The AUDPC for lesion lengths was calculated in control and inoculated CO447 and CO450 corn lines. Lesion size was measured on CO450 and CO447 leaves from day 2 to day 10 after inoculation with CMN14-5-1. Asterisks denote significant differences, * *p* < 0.05. (**C**) Progression of disease severity in CO447 and CO450 lines from 2 to 10 dpi was calculated using a disease severity scale of 0–5. The values represent the AUDPC values mean of 6 biological replicates and standard error was represented over time.

**Figure 2 plants-12-01475-f002:**
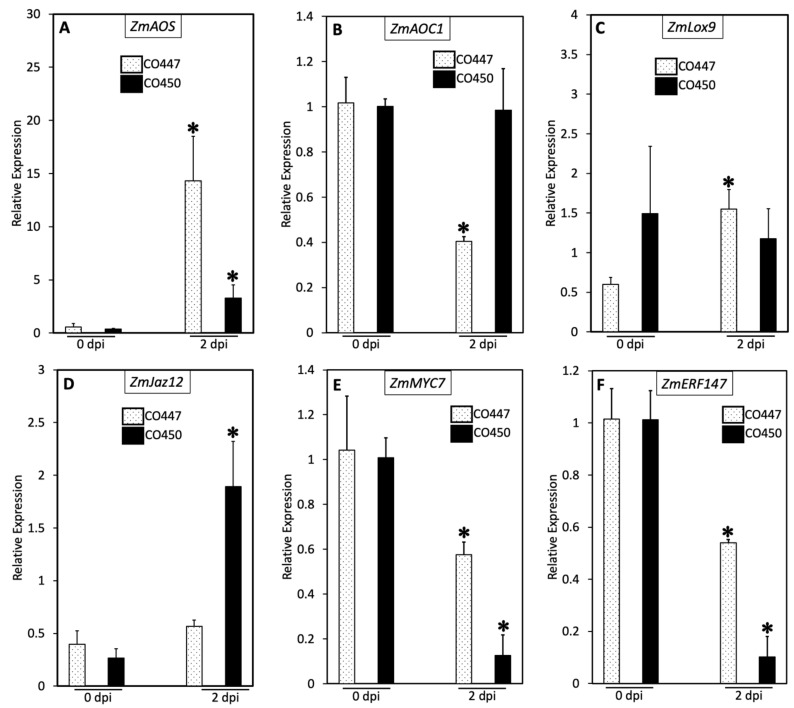
Induced resistance against Goss’s wilt disease in corn plants is not associated with the JA-dependent pathway. RT–qPCR analysis showing the transcript levels of the JA-regulated genes *ZmAOS* (**A**), *ZmAOC1* (**B**), *ZmLox9* (**C**), *ZmJaz12* (**D**), *ZmMYC7* (**E**), and *ZMERF147* (**F**) in response to CMN14-5-1 of CO447 and CO450 corn lines. Leaves were sampled at 0 h and 48 h after treatments. Reference genes were Elongation Factor 1 (*EF1a*) for (**A**,**C**,**D**) and Actin for (**B**,**E**,**F**). The results are representative of three biological replicates. The error bars represent the standard error. Asterisks denote significant differences; * *p* < 0.05.

**Figure 3 plants-12-01475-f003:**
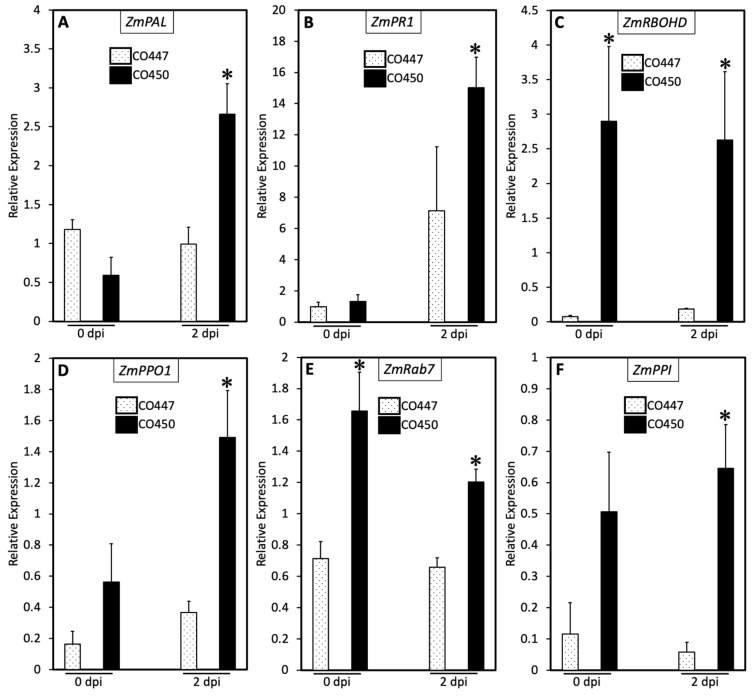
SA and PCD regulate disease resistance in corn plants in response to *C. nebraskensis.* Real time RT–qPCR analysis showing the transcript levels of the SA-regulated genes *ZmPAL* (**A**), *ZmPR-1* (**B**), and the PCD marker genes *ZmRBOHD* (**C**), *ZmPPO1* (**D**), *ZmRab7* (**E**), and *ZmPPI* (**F**) in response to CMN14-5-1 of the CO447 and CO450 corn lines measured. Leaves were sampled at 0 h and 48 h after treatments. Reference gene was the Elongation Factor 1 (*EF1a*). The results are representative of three biological replicates. The error bars represent the standard error. Asterisks denote significant differences, * *p* < 0.05.

**Figure 4 plants-12-01475-f004:**
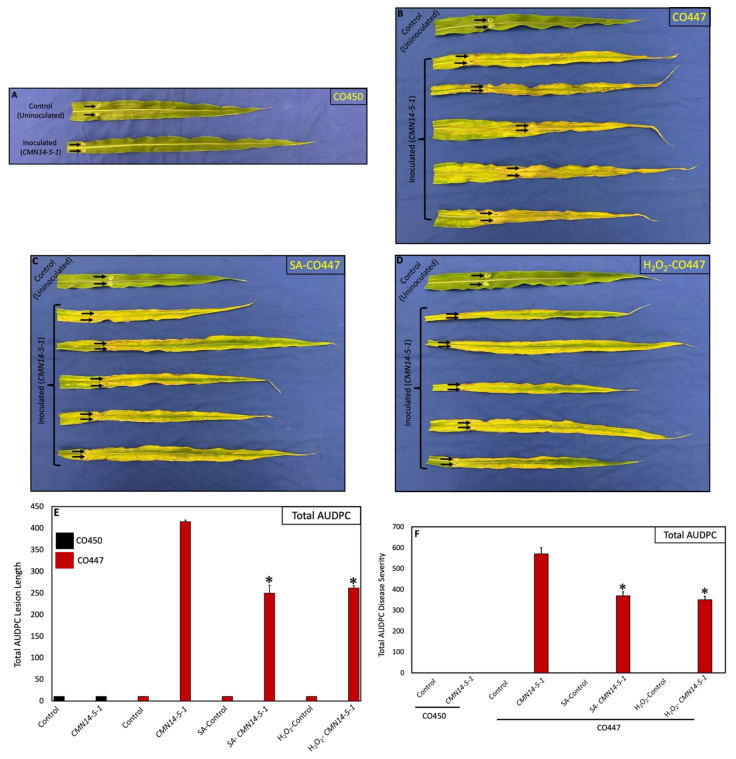
SA and H_2_O_2_ confer partial disease resistance to *C. nebraskensis* (CMN14-5-1) in the susceptible CO447 corn line. (**A**–**D**) Phenotype of the tested corn plants. Wounded tolerant CO450 (**A**) and susceptible CO447 (**B**). Plants were pretreated with either the PPB buffer (control) or the aggressive CMN14-5-1. Control leaves were treated with 20 μL of buffer, and inoculated leaves were inoculated with the bacterial suspension. Arrows point to the inoculation site. Local defense response in the susceptible CO447 plants pretreated with SA (**C**) or H_2_O_2_ (**D**) (500 μM each). The tested plants were inoculated with CMN14-5-1 48 h after local treatments. (**E**) The AUDPC for lesion lengths was calculated for control and inoculated CO447 plants pretreated with SA or H_2_O_2_. (**F**) Disease severity progression in CO447 plants pretreated with SA or H_2_O_2_. The values are the average of the AUDPC values for 12 biological replicates at 7 and 14 days after inoculation, based on a disease severity scale of 0 to 6. The error bars represent the standard error. Asterisks denote significant differences of the treated inoculated CO447 plants relative to the inoculated CO447 plants; * *p* < 0.05.

## Data Availability

The data presented in this study are available on request from the corresponding author. The data are not publicly available due to privacy concerns and connections to other ongoing studies.

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
