# Peer review of "Goss’s Wilt Resistance in Corn Is Mediated via Salicylic Acid and Programmed Cell Death but Not Jasmonic Acid Pathways"

_plants, 2023, doi:10.3390/plants12071475_

Round 1

Reviewer 1 Report

The authors studied the role of SA and PCD defense-related genes in the interactions between corn and Goss’s wilt caused bacterial pathogen Clavibacter nebraskensis. Overall, the study is well structured, the experiments made correctly.  The results are informative and will be of interest to the readers.

Some minor text editing is required (for example: Line 3. Change pathways to Pathways; Lines 79 and 80. Use SA and JA without define).

Best wishes

Reviewer 2 Report

Goss’s wilt caused by the bacterial pathogen Clavibacter nebraskensis is an important issue in maize production worldwide. While it has been demonstrated that SA and JA pathways might play opposite roles in resistance against other pathogens, their roles in maize defense against Goss’s wilt have not been fully understood. This work examined the disease phenotypes to Goss’s wilt using a pair of lines with contrast resistance/susceptibility phenotypes. Furthermore, the authors investigated the expression patterns of several genes associated with SA and JA pathways, as well as that related to programmed cell death (PCD), and found that SA pathway and PCD genes are upregulated in resistance line whereas JA pathway genes are only induced in susceptible line. Furthermore, exogenous application of SA enhanced the resistance to Goss’s wilt. Thus, they concluded that maize resistance to Goss’s wilt is mainly dependent on SA and PCD pathway but not JA pathway.

However, the evidences provided in this study are not sufficient to support the conclusion drawn from the data available. First, this work lacks genetic evidences by showing SA- and JA-pathway mutants are oppositely compromised in the disease resistance. Second, the study did not show whether exogenous JA or analogs could not enhance or even reduce the resistance to Goss’s wilt cause. Third, the genes selected for JA pathways are not typical, as 1) there are three ZmAOS genes in maize genome, but only one ZmAOS gene was examined; 2) the maize LOX genes directly associated with JA biosynthesis are ZmLOX7 and ZmLOX8, rather than ZmLOX9.

Minor points:

Abstract: line 21: suggest change “overexpressed” to “was induced”.

Zm ID for selected genes are lacking;

Reviewer 3 Report

Plant-pathogen interaction is complex and understanding the underlying molecular mechanisms is fundamental for devising novel crop protection strategies. Shumilak et al. investigated maize responses to the pathogenic bacteria Clavibacter nebraskensis; gene expression analysis of two maize genotypes indicated that salicylic acid (SA) and reactive oxygen species (ROS) could be involved in plant immunity to this pathogen, which was then confirmed by chemical treatments.

This is well-written and interesting manuscript; there are however some aspects the author may consider to enhance the manuscript:

1. The authors analyze several genes involved in the SA and ROS signaling, and also jasmonic acid (JA). The rationale behind the choice of JA marker genes shown in Fig. 2 is not entirely clear; I would suggest to better explain it in the introduction (perhaps moving information already present in the discussion).

2. Fig. 1.

- Please clarify the time point(s) used for collecting the data shown. Specifically, panel A mentions data were collected from 5 to 14 dpi, panel C from 2 to 10 dpi, and no information is provided for panel B.

- Panel B includes an asterisk, but no information is provided about its meaning.

- Panel C caption indicates "standard error over time", but the plot shows no error bars. It would be further useful to use the same plot to present statistics. Specifically, if differences shown in 1C at 2 dpi are significant, they could be used to claim that immune response of CO450 is faster than CO447.

3. Results of Fig. 4 indicate that SA alone is not sufficient to confer full resistance. To promote robust plant immunity, recent work showed that SA acts in concert with abscisic acid (Plant Commun 2020;1(5):100099 https://doi.org/gpmxmc ) and pipecolic acid (Mol Plant Microbe Interact 2019;32(10):1303-1313 https://doi.org/jxbq ). Can the authors present experimental data to clarify this aspect or at least discuss it?

4. Student’s t-test is commonly used for comparisons of two groups; the authors however report the use of "Tukey test", which is a post hoc test often used after ANOVA test. Additionally, figure captions report "Asterisks indicate significant differences, * p < 0.05", however it is not clear which conditions and if two or more groups were compared. Can the authors clarify the statistical analysis part and information presented in the figures?

5. Please clarify providers and catalog numbers of the reagents used, especially for those used in the chemical treatment assays.

Round 2

Reviewer 2 Report

The revision did not solve the problems that the data presented from non-representative JA-biosynthesis genes are not sufficient to exclude the possibility of JA not involved in the resistance to Goss's bacterial wilt. Additional data for qRT-PCR examination of other AOSs, LOX7 and LOX8 are required at least to support the conclusion.

Round 3

Reviewer 2 Report

The authors deployed several alternative genes instead of ZmAOS, ZmLOX7, and ZmLOX8 for JA pathway to affirm the conclusion that JA pathway is not involved in the resistance to Goss's wilt. While I do not have problem with the data, I would suggest the authors to be very careful on interpretation of those new data, since ZmLOX1, ZmLOX3 and ZmLOX5 are belonging to 9-LOX branch, which are not involved in JA biosynthesis (13-LOX branch). I would suggest the authors to delete these three genes.

Author Response

Our sincere thanks to the reviewer for their suggestion.

We have made the changes as suggested.

Thank you again.